# Effectiveness of food supplement on treatment outcomes and quality of life in pulmonary tuberculosis: Phased implementation approach

Amarendra Mahapatra[1], Kannan Thiruvengadam[2], Dina Nair[2], Chandrasekaran Padmapriyadarsini[2]*, Beena Thomas[2], Sanghamitra Pati[1], Gandham Bulliyya[1], Dasarathi Das[1], Jayeeta Chowdhury[3], Anand Bang[3], Soumya Swaminathan[4]

1 ICMR-Regional Medical Research Centre, Bhubaneswar, Odisha, India, 2 ICMR-National Institute for Research in Tuberculosis, Chennai, India, 3 Tata Trusts, Mumbai, India, 4 Indian Council of Medical Research, New Delhi, India

* pcorchids@gmail.com

**Data Availability Statement:** As the data set has sensitive information from the study participants, data is not freely available. Upon request, only

## Abstract

### Background

By encouraging treatment adherence and lowering mortality, dietary supplements can serve as adjuvant therapy for the success of medical interventions. We determined the effect of locally accessible food supplements on treatment outcomes, and health-related quality of life in patients with pulmonary tuberculosis initiating anti-tuberculosis treatment (ATT) in Odisha, India.

### Method

Between September 2017 and December 2018, implementation research in patients with newly diagnosed sputum smear-positive pulmonary tuberculosis initiating ATT in five districts of the tribal belt of Odisha, offered food supplements along with ATT in a phased manner. Clinical symptoms, anthropometry, sputum for M. tuberculosis (M. tb), health-related quality of life and return to normal function were assessed periodically, and favourable treatment outcome (cure or treatment completed) was measured at the end of treatment. The effect of the food supplement on unfavorable outcomes (treatment failure, death, or lost-to-follow-up) was modelled using mixed-effects Poisson regression to determine the risk factors.

### Results

Among the 761 participants enrolled, 614 participants received the food supplement and 147 did not receive the food supplement. Among the 614 participants in the supplement group, 537 (87%) had a favorable outcome and among the 147 participants in the no-supplement group, 113 (77%) had a favorable outcome (p = 0.0017). Higher age (>55 years) [aRR = 2.1(95% CI: 1.1–3.8)], male gender [aRR = 1.7(95% CI: 1.2–2.9)], and smear

deidentified data is available after obtaining permission from the RMRC & NIRT Institutional Ethics Committee Chair (contact via Ethics committee member secretary) for researchers who meet the criteria for access to confidential data. The email IDs of the Member Secretary of the Institutional Ethics committees are: RMRC, Bhubaneswar: Dr.Sidhartha Giri: sidhartha.g@icmr. gov.in NIRT, Chennai: Dr.G.Narendran: gopalannaren@yahoo.co.in.

**Funding:** The study was supported by Tata Trusts. The funders had no role in study design, data collection and analysis, decision to publish, or preparation of the manuscript. The funder provided salary support to two authors (JC & AB) as employees of Tata Trusts. This does not alter our adherence to PLoS One policy on sharing data and material upon request.

**Competing interests:** JC & AB are employees of TATA TRUSTS. Rest of the authors have declared that no competing interests exist.

grading $\geq$2+ [aRR = 1.5 (95% CI: 1.1–2.2)] were associated with unfavorable treatment outcomes. Nutritional status, quality of life and lung health showed significant improvement from baseline in the supplement group.

## Conclusion

Improvement in the nutritional status of the patient can be considered a predictor of treatment success rates. Early food supplementation has a positive impact on the nutritional status.

## Introduction

Tuberculosis (TB) remains one of the major infectious causes of morbidity and mortality worldwide [1]. Undernutrition is an important risk factor as well as a common consequence of TB. It is associated with an increased risk of mortality and poor treatment outcome probably due to decreased drug absorption or weakened cell-mediated and humoral immunity and decreased phagocytic functions [2–6]. Undernourishment with low body mass index (lower than 18.5kg/m$^2$), and lack of adequate weight gain with TB treatment are associated with longer time to sputum conversion, higher risk of hepatotoxicity, relapse, and mortality [7–10]. A large study from rural India showed that 80% of women and 67% of men had moderate to severe undernutrition and body weights did not return to normal even at the end of treatment [11].

Food supplements can act as an adjunct for treatment success conferring socioeconomic and survival benefits to patients with TB. Patients who received food supplements during the initial phase of TB treatment gain more weight, have a shorter time to sputum conversion, high cure rate as well as decreased mortality [12–15]. At the population level, undernutrition and poor weight gain with treatment are thus important modifiable risk factor for TB.

Though India has the highest burden of TB and undernutrition, strategies to overcome this interlinked epidemic need to be strengthened and prioritized. We aimed to determine the effect of locally available food supplements on treatment outcomes and the nutritional status of the patients initiated on anti-TB treatment in five districts of Odisha. We also assessed the Quality of Life, lung health, and return to normal functionality among patients receiving food supplements along with anti-TB treatment.

## Methods

### Study setting and design

The study was conducted between September 2017 to December 2018 in the 19 Designated Microscopy Centres (DMC) in five districts of Odisha namely Gajapati (Adava, Chandragiri, Gumma, Kashinagar, Khajuripada, Khandawa, Mohana, R. Udayagiri, and Rayagada), Kalahandi (Biswanathpur and Lanjigarh), Kandhamal (Phiringia and Phulbani), Malkangiri (Mathili and Pandripani) and Rayagada (Bissamkatak, Durgi, and Muniguda).

The design of the study was a stepped-wedge that offered a logical as well as ethical feasibility [16]. This method was used as it was not feasible to provide the intervention to every individual/community at once. The order in which the different individuals or clusters receive the intervention was determined at random and, it was ensured that by the end of the random allocation, all individuals or groups received the intervention.

| Phase | District (Cluster) | DMC/ Strata | Time points | | | | | | | Follow up (6 months) |
|---|---|---|---|---|---|---|---|---|---|---|
| | | | T0 | T1 | T2 | T3 | T4 | T5 | T6 | |
| 1 | Gajapati | Chandragiri | 1 | 16 | 19 | 20 | 34 | 27 | 22 | |
| | | Gumma | | | | | | | | |
| | | Mohana | | | | | | | | |
| | | Rayagada | | | | | | | | |
| 2 | Gajapati | Adava | 1 | 10 | 19 | 21 | 23 | 27 | 20 | |
| | | Kashinagar | | | | | | | | |
| | | Khajuripada | | | | | | | | |
| 3 | Gajapati | R.udayagiri | 2 | 9 | 9 | 22 | 26 | 25 | 23 | |
| | Malkangiri | Mathili | | | | | | | | |
| | | Pandripani | | | | | | | | |
| 4 | Kalahandi | Biswanathpur | 2 | 10 | 9 | 6 | 19 | 23 | 22 | |
| | | Lanjigarh | | | | | | | | |
| | Gajapati | Khandawa | | | | | | | | |
| 5 | Kandhamal | Phiringia | 2 | 9 | 10 | 7 | 9 | 24 | 21 | |
| | | Phulbani | | | | | | | | |
| | Rayagada | Bissamkatak | | | | | | | | |
| 6 | Rayagada | Muniguda | 2 | 10 | 11 | 12 | 7 | 9 | 24 | |
| | | Ambadola | | | | | | | | |
| | | Durgi | | | | | | | | |

T0 – Baseline; T1 – T6 – Supplement implementation; Follow-up – post enrolment follow-up (6 months);
Duration in the study – 6 months; Interval between T1 to T2 - 2 months;
Highlighted region indicates the implementation of the Intervention and the numbers indicates the enrolment

**Fig 1. Stepped-wedge cluster selection in 19 designated microscopy centers in study districts of Odisha.**

Thus, a phased implementation of food supplements was sequentially introduced in all the selected clusters. There were six clusters consisting of three DMCs in five clusters and four DMCs in one cluster. The supplement was initiated in one cluster where the participants enrolled were given the supplement in addition to the anti-TB treatment. The participants in the other five clusters continued anti-TB treatment without the supplement. After every two months, the next cluster was selected and the participants enrolled were given the supplement. The enrolment stopped when the last cluster had enrolled participants and the supplement was given for two months. The outcome parameters were measured on all the participants in all the clusters at every second, fourth, and sixth-month follow-up. The logistics of the stepped-wedge trial design used are shown in Fig 1.

## Study population

The participants were newly diagnosed with sputum smear-positive pulmonary TB initiated on anti-TB treatment. All the participants in a cluster before the introduction of food supplements in that cluster were considered as no supplement group. Participants who were allergic to pulses/nuts and not willing to provide informed consent were excluded. All participants were followed up for a minimum period of six months post-supplementation.

## Study procedures

A baseline assessment of all participants was done and a detailed clinical and demographic history was collected. Anthropometric measurements such as body weight, body mass index

(BMI), mid-upper arm circumference (MUAC), and waist circumference were recorded. A retrospective 24-hour dietary recall and dietary diversity score was assessed by a trained study staff. Nutrient intake was calculated using "Digest" software, a specially designed software package to analyze South Indian diets. Sputum results and treatment details were collected from lab registers and treatment cards at pre-determined follow-up time points available at the DMCs. Participants were counseled for regular intake of drugs as well as supplement.

The composition of the food supplement provided included Rice (20 kgs), or Ragi (10kgs), Local Arhar dhal (Kandol) (9 kgs), Mustard Oil (2 kgs) along with 500gms of "Sathu" (flour made from groundnut, wheat, flat rice, and chickpea which contained the following—protein (19 gm), fat (6 gm), carbohydrates (71gm), calcium (200 mg), energy (413kcal), Vit A (300 IU), Vit C (300 mg) and Vitamin D (100 IU) (Box 1). The quantity of food supplement provided supported a family of five for one month and cost Rs.1500/- per month. This was intended for the index patient with TB, while additional supply was given for the family members of the patient. The type of pulses and oil were added based on local practices and preferences. It was supplied to the index patient as fortnightly packs at the Directly observed treatment short course (DOTs) Center by the study staff till the end of his /her treatment period. Nutritional counseling was provided to the participant. The participants were followed up every month with clinical exams, anthropometry, dietary recall, and drug adherence. Sputum results were collected from the records at DMCs. Adherence to anti-TB treatment and supplement was ensured by surprise home visits and requisitioning of patients during monthly visits. The health-related Quality of Life (HRQOL), lung health, and return to normal

### Box 1. Composition of the food supplement provided to the participants

| Food category | Food item | Quantity (g/day)/ RDA | Energy (Kcals) | Protein (g) | Fat (g) | Carbo-hydrates (g) | Others |
|---|---|---|---|---|---|---|---|
| Grains | Rice | 100 | 162 | 3.17 | 1.28 | 33.17 | Calcium 12 mg -1% Iron-1.4 mg-8% Potassium-42 mg-1% Vitamin A-10 ug 1% Dietary fibre -67g |
| | Ragi (Millet) | 100 | 344 | 7.3 | 1.3 | 72.0 | Folic Acid-18.3ug Carotene- 42ug |
| Pulse | Local Arhar dhal Kandol) | 40 | 360 | 20.8 | 5.6 | 59.8 | Calcium -56 mg Phosphorus- 331 mg Iron -5.3 mg |
| Fat | Mustard Oil | 20 | 900 | – | 100 | – | – |
| | Sathu* | 50 | 413 | 19 | 6 | 71 | Vit A- 300 IU Vit C- 300 mg Vit D- 100 IU Calcium -200 mg |

*RUCHI Brand (local)—Flour made from groundnut, wheat, flat rice, and chickpea
Ref: C.Gopalan et.al, Nutritive value of Indian Foods, ICMR- NIN.

functionality among patients receiving food supplement and among patients not receiving supplement was assessed at baseline, second month, and sixth-month WHOQOL-BREF scale questionnaire and St. George's respiratory questionnaire (SGRQ).

The outcome measures were the response to treatment in terms of favorable and unfavorable outcomes at the end of treatment in both groups (supplement and no supplement group). The factors associated with death and loss of follow-up were also determined. The effect of food supplementation on nutritional status was also evaluated. The favorable outcome included treatment completed and cured. The unfavorable treatment outcome included death during treatment, loss to treatment follow-up, and treatment failure.

## Sample size

The treatment success rate of new smear-positive cases in the Gajapati district was 73%, as compared to the national average of 90% [17]. In the study area, 19 DMCs recorded 285 new smear-positive cases (NSP) per quarter (approximately 1140 per year and an average of 5 cases per month in each DMC). To achieve a treatment success rate of 90% with the food supplement by improving the adherence rate, assuming an intra-cluster correlation of 0.05, [18], 95% confidence level, 90% power, and 20% of refusals/loss to follow-up, the required sample size was 703 patients over one year. The sequence of units (clusters) that initiated the food supplement at each period was determined randomly.

## Data collection

The data capture was done offline using an Android handheld device after a range of logic checks. The structured case reporting forms were designed in the Epi Info software package version 7.2.1.0 (Centers for Disease Control, Atlanta, GA, USA). A quality check was performed periodically for accuracy and completeness of the data and to minimize missing data.

## Statistical considerations

The data were analyzed using STATA software version 16.0 (Stata Corp, Texas, USA). Categorical and continuous variables are summarized as proportions and medians with interquartile range (IQR), respectively. The Mann-Whitney U test was used to compare the median between the groups. Proportions were compared using the z-proportion test. Fisher's exact test was used to compare the baseline factors among the supplement and no-supplement groups. Univariable and multivariable mixed-effects Poisson regression with random effects and person-time as offset was used to measure the association of various factors with unfavorable outcomes (treatment failure, loss to follow–up and mortality) with an adjustment to the potential factors and time-varying covariates. The effect of the food supplementation on unfavorable outcomes was modelled using mixed-effects Poisson regression with random cluster effects allowing the inclusion of risk factors such as age, gender, body mass index, diabetes status, the habit of alcohol and smoking, type of cooking fuel and smear grading. These models were adjusted for clustering and time effects. The TB treatment outcome and nutrition factors were chosen based on literature review and data availability. Regression models were then conducted, taking into account these covariates. A p-value of less than 0.05 was considered statistically significant for all analyses.

The study was approved by the Institutional Ethics Committees of the Regional Medical Research Centre, Bhubaneswar.

## Results

A total of five districts and 19 DMCs were included in the study. There were a total of 761 participants, among whom 614 participants received the food supplement and 147 did not receive the food supplement. The number of DMCs and the number of participants enrolled in each phase from September 2019 to July 2018 is shown in Fig 1.

### Clinical and demographic details

The socio-demographic and clinical characteristics of the participants are shown in Table 1. Among the 761 participants with TB, 74% were males and 54% were extremely underweight (Body mass index (BMI) <16.5 kg/m$^2$). Among those with alcohol consumption, a higher proportion were in the supplement group. Similarly, those with higher bacillary load, a larger proportion were in this group.

### Treatment outcomes

Among the 614 participants in the supplement group, 537 (87%) had a favorable outcome and among the 147 participants in the no-supplement group, 113 (77%) had a favorable outcome (p = 0.0017). The incidence of failure and loss for follow-up was significantly higher in the group who did not receive the supplement [aIRR = 9.73 (95% CI: 2.5–37.8) and aIRR = 2.2 (95% CI: 1.3–3.6)] (Table 2).

### Factors associated with unfavourable treatment outcomes

Multivariable analysis showed that age more than 55 [aRR = 2.1(95% CI: 1.1–3.8)], male gender [aRR = 1.7(95% CI: 1.1–2.9)], and smear grading ≥2+ [aRR = 1.5(95% CI: 1.1–2.2)] were independently associated with a higher incidence of unfavourable treatment outcomes (Table 3). Male gender [aRR = 2.1 (95% CI: 1.1–4.2)] was independently associated with increased loss to follow-up (Fig 2A and 2B). Age more than 55 years [aRR = 3.9 (95% CI: 1.2–12.5)], and BMI<16.5 [aRR = 8.8(95% CI: 1.2–66.2)] was found to be associated with increased risk of mortality among the participants (Fig 2C). Adding the group with change in the BMI as an interaction term showed that the supplement without any weight gain [aRR = 2.4(95% CI: 1.5–3.7)] and no supplement without weight gain [aRR = 3.7(95% CI: 2.3–5.8)] were significantly having an increased risk of unfavourable treatment outcomes. However, there is an increased risk of loss to follow-up among the no supplement without weight gain [aRR = 3.2 (95% CI: 1.9–5.5)].

### Nutritional status and food supplementation

Mean and standard deviation of various anthropometric measurements like weight, waist circumference, BMI, and MUAC were calculated for both supplement and no supplement groups. The change and the increase from baseline (%) were calculated for the above four indices. The changes from the baseline were significantly higher in the supplement group in terms of weight, BMI, and waist circumferences (p<0.001). MUAC changes were not significantly different from the baseline in the supplement group (Fig 3).

The median score in the WHO-QOL BREF scale was significantly higher in the supplement group at month 2 and month 6 visits in the physical, psychological, and social relationship domains. Assessment of lung health by the SGRQ scale showed a significantly higher score in the no-supplement group (Table 4).

**Table 1. Baseline characteristics of the participants enrolled in the study (n = 761).**

| Characteristics | No supplement (N = 147) | Supplement (N = 614) | p-value |
|---|---|---|---|
| | N (%) | N (%) | |
| **Age in years** | | | |
| 18–30 | 29 (20) | 185 (30) | 0.066 |
| 31–45 | 51 (35) | 184 (30) | |
| 46–55 | 29 (20) | 121 (20) | |
| >55 | 38 (26) | 124 (20) | |
| **Gender** | | | |
| Female | 33 (22) | 167 (27) | 0.240 |
| Male | 114 (78) | 447 (73) | |
| **Baseline Weight in kg** [median (IQR)] | 40 (35.2–43.9) | 40.3 (35.5–45.1) | 0.398 |
| **Change in Weight at 2nd month** | | | |
| ≥5.0% | 27 (18%) | 216 (35%) | **<0.001** |
| 0.0% - 5.0% | 20 (14%) | 214 (35%) | |
| <0.0% | 43 (29%) | 94 (15%) | |
| NA* | 57 (39%) | 90 (15%) | |
| **Body mass index (BMI)** | 16.4 (15.1–17.9) | 16.2 (14.9–17.8) | 0.687 |
| **BMI (kg/m²) *** | | | |
| Normal [18.5–24.9] | 27 (18) | 96 (16) | 0.413 |
| Underweight [16.5–18.4] | 37 (25) | 187 (30) | |
| Extremely Underweight [<16.5] | 83 (56) | 330 (54) | |
| **Change in BMI (at 2nd month)** | | | |
| ≥5.0% | 22 (15%) | 224 (36%) | **<0.001** |
| 0.0% - 5.0% | 18 (12%) | 201 (33%) | |
| <0.0% | 50 (34%) | 99 (16%) | |
| NA** | 57 (39%) | 90 (15%) | |
| **Mid-Upper Arm Circumference** | 22 (20–23) | 20.5 (19–22.4) | **0.001** |
| **Waist Circumference** | 66 (64–68) | 66 (64–69) | 0.207 |
| **Diabetes** | | | |
| No | 143 (97) | 600 (98) | 0.752 |
| Yes | 4 (3) | 14 (2) | |
| **Hypertension** | | | |
| No | 146 (99) | 611 (100) | 0.773 |
| Yes | 1 (1) | 3 (0) | |
| **Alcohol consumption** | | | |
| No | 106 (72) | 383 (62) | **0.027** |
| Yes | 41 (28) | 231 (38) | |
| **Smoking habit** | | | |
| No | 139 (95) | 554 (90) | 0.098 |
| Yes | 8 (5) | 60 (10) | |
| **Income (000's) INR/per Annum** | 45 (30–80) | 45 (30–60) | 0.359 |
| **Sputum smear grading** | | | |
| <2+ acid fast bacilli | 64 (44) | 330 (54) | **0.026** |
| ≥2+ acid fast bacilli | 83 (56) | 284 (46) | |
| **Fuel used for cooking** | | | |
| Smoke | 138 (94) | 541 (88) | **0.043** |

(*Continued*)

**Table 1.** (Continued)

| Characteristics | No supplement (N = 147) | Supplement (N = 614) | p-value |
|---|---|---|---|
| | N (%) | N (%) | |
| Smokeless | 9 (6) | 73 (12) | |

BMI-Body Mass Index

Values are given as median (First and Third Quartiles) and Frequency (Percentages)

Fisher's Exact and Mann-Whitney U test was used at a 5% level of significance

* Reference: Guidance Document—Nutritional Care & Support for TB patients in India 2017.

NA** indicates that it was not included in the calculation of the p-value.

## Discussions

Nutritional support is an essential component to improve treatment success rates and reduce mortality in patients with active TB disease and is consistent with the patient-centered care (pillar 1) of the end TB strategy.

A recent study done among active patients with TB in India concluded that nutrition supplements and regular counselling can help patients to gain weight leading to improved treatment outcomes [19]. Our study also confirms the role of food supplements in improving adherence thereby increasing the treatment completion rates.

**Table 2. Comparison of the effect of food supplement on treatment outcomes in participants enrolled in the trial using various models (n = 761).**

| | Total (n = 761) | Fav. (n = 650) | Unfavorable (n = 111) | | | |
|---|---|---|---|---|---|---|
| | | | Composite (n = 111) | Failure (n = 10) | Mortality (n = 31) | LTFUP (n = 70) |
| **Groups** | | | | | | |
| Supplement | 614 | 537 (87) | 77 (13) | 3 (0.5) | 28 (5) | 46 (7) |
| No Supplement | 147 | 113 (77) | 34 (23) | 7 (5) | 3 (2) | 24 (16) |
| **Model—A: Unadjusted Model (IRR [95% CI])** | | | | | | |
| IRR of No Supplement group with reference to Supplement group | | | 1.9 (1.3–2.9) | 10.9 (2.8–42.4) | 0.5 (0.2–1.7) | 2.4 (1.4–3.9) |
| | | | **<0.001** | **<0.001** | 0.299 | **<0.001** |
| **Model—B: Individual Adjusted Model (aIRR [95% CI])** | | | | | | |
| IRR of No Supplement group with reference to Supplement group | | | 1.8 (1.2–2.8) | 9.7 (2.5–37.8) | 0.6 (0.2–1.9) | 2.2 (1.3–3. 6) |
| | | | **0.004** | **0.001** | 0.353 | **0.003** |
| **Model—C: Cluster adjusted model (IRR [95% CI])** | | | | | | |
| IRR of No Supplement group with reference to Supplement group | | | 1.6 (1.2–2.2) | 10.43 (2.5–43.8) | 1.0 (0.4–2.5) | 1.6 (1.1–2.3) |
| | | | **0.004** | **0.001** | 0.914 | **0.010** |
| **Model—D: Hussey and Hughes Time-adjusted model (IRR [95% CI])** | | | | | | |
| IRR of No Supplement group with reference to Supplement group | | | 1.5 (1.1–2.3) | 6.7 (1.6–28.3) | 0.9 (0.4–2.1) | 1.6 (1.1–2.2) |
| | | | **0.015** | **0.009** | 0.844 | **0.016** |

Fav.–Favourable

Model-A: An individual level, Unadjusted (IRR—Incidence Rate Ratio) effect of food supplement was estimated using multilevel mixed effects Poisson regression with person-time as an offset.

Model-B: An individual level, adjusted (IRR—Incidence Rate Ratio) effect of Supplement was estimated using multilevel mixed effects Poisson regression with person-time as an offset. This model was adjusted for the covariates such as age, gender, body mass index, diabetes status, habit of alcohol and smoking, type of cooking fuel, smear grading at the baseline

Model-C: This model estimated the effect of the naïve Supplement i.e., a beneficial effect after adjusting for clustering effect without adjusting for time using multilevel mixed effects Poisson regression with person-time as offset

Model-D: This model estimated the effect of Supplement adjusting for both the time and clustering effect using multilevel mixed effects Poisson regression with person-time as offset

**Table 3. Factors associated with unfavorable TB treatment outcomes in participants enrolled in the study (n = 761).**

| Characteristics | Fav (N = 650) | Unfav (n = 111) | RR [95% CI] | p-value | aRR [95% CI] | p-value |
|---|---|---|---|---|---|---|
| | N (%) | N (%) | | | | |
| **Group** | | | | | | |
| Supplement | 537(87) | 77(13) | 1.0 | | 1.0 | |
| No Supplement | 113(77) | 34(23) | 1.9(1.3–2.9) | **<0.001** | 1.8(1.2–2.7) | 0.005 |
| **Age in years** | | | | | | |
| 18–30 | 195(91) | 19(9) | 1.0 | | 1.0 | |
| 31–45 | 198(84) | 37(16) | 1.9(1.1–3.3) | **0.025** | 1.6(0.9–2.9) | 0.085 |
| 46–55 | 127(85) | 23(15) | 1.9(1.0–3.5) | **0.038** | 1.6(0.9–3.0) | 0.141 |
| >55 | 130(80) | 32(20) | 2.4(1.4–4.3) | **0.002** | 2.1(1.1–3.8) | **0.016** |
| **Gender** | | | | | | |
| Female | 181(90) | 19(10) | 1.0 | | 1.0 | |
| Male | 469(84) | 92(16) | 1.8(1.1–2.9) | **0.022** | 1.7(1.1–2.9) | 0.043 |
| **Body Mass Index (kg/m$^2$)** | | | | | | |
| ≥18.5 | 110(89) | 14(11) | 1.0 | | 1.0 | |
| 16.5–18.4 | 196(87) | 28(13) | 1.1(0.6–2.1) | 0.789 | 1.2(0.6–2.3) | 0.569 |
| <16.5 | 344(83) | 69(17) | 1.6(0.9–2.8) | 0.131 | 1.8(0.9–3.2) | 0.060 |
| **Diabetes** | | | | | | |
| No | 633(85) | 110(15) | 1.0 | | 1.0 | |
| Yes | 17(94) | 1(6) | 0.3(0.1–2.5) | 0.295 | 0.3(0.1–2.1) | 0.216 |
| **Alcohol consumption and Smoking** | | | | | | |
| Nil | 418(87) | 62(13) | 1.0 | | 1.0 | |
| Either one | 183(82) | 39(18) | 1.3(0.9–1.9) | 0.229 | 1.1(0.7–1.6) | 0.747 |
| Both | 49(83) | 10(17) | 1.2(0.6–2.4) | 0.554 | 1.0(0.5–2.1) | 0.973 |
| **Type of fuel used for cooking** | | | | | | |
| Smokeless | 74(90) | 8(10) | 1.0 | | 1.0 | |
| Smoke | 576(85) | 103(15) | 1.6(0.8–3.3) | 0.192 | 1.3(0.6–2.7) | 0.510 |
| **Sputum Smear Grading** | | | | | | |
| <2+ | 349(89) | 45(11) | 1.0 | | 1.0 | |
| ≥2+ | 301(82) | 66(18) | 1.5(1.1–2.3) | **0.023** | 1.5(1.1–2.2) | 0.049 |

Fav.–Favourable; Unfav.–Unfavourable

Smokeless—Gas Cylinder/Electricity; Smoke—Fire Wood/Charcoal/Kerosene

Values are given as Frequency (Percentages), and Person years were used as an offset variable in the estimation IRR (Incidence Rate Ratio)

Improvement in nutritional status and weight gain are important predictors of favourable treatment outcomes in patients with TB [20, 21]. A weight gain of less than 5% in the initial months of treatment is a risk factor for relapse [6]. Food supplements as an adjunct to TB treatment increased lean mass and physical function [13]. A recent study done in Ethiopia demonstrated that being underweight before initiation and after two months of treatment was associated with unfavourable treatment outcomes [22]. In our study the patients in the supplement group had a significant weight gain and improvement in body mass index and mid-arm circumference compared to the patients in the control arm with anti-TB treatment alone.

A few studies done to evaluate the role of food supplements on TB treatment outcomes found no direct benefit [13, 23, 24]. Higher success rates in the supplement group of this study are associated with the food supplement provided and subsequent improvement in nutritional status.

A randomized controlled trial done in Timor-Leste suggested that food incentives have little effect on adherence and completion of treatment though there was weight gain and early

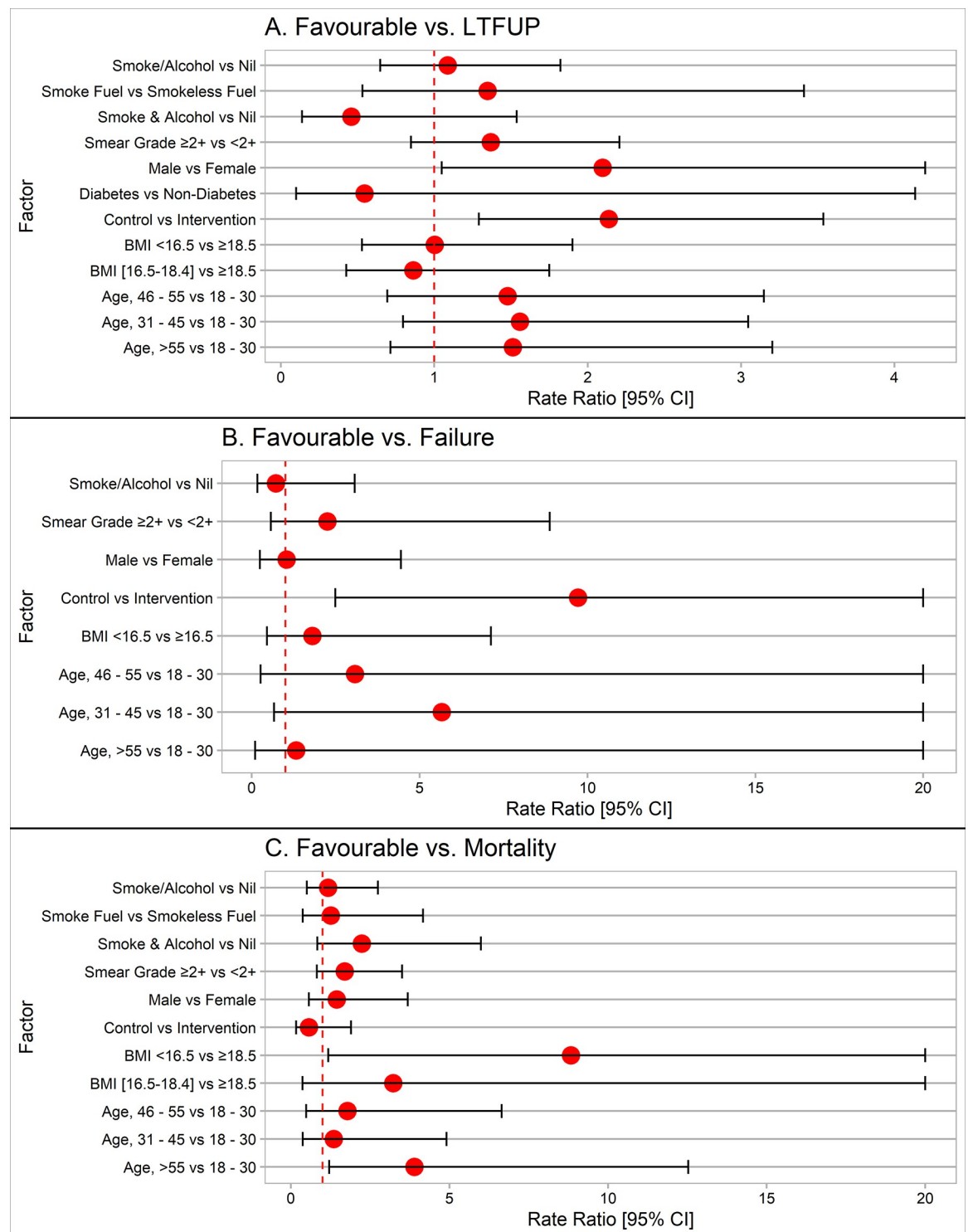

**Fig 2.** Risk factors associated with loss to follow-up (A) / Failure (B) / Mortality (C) among the study participants.

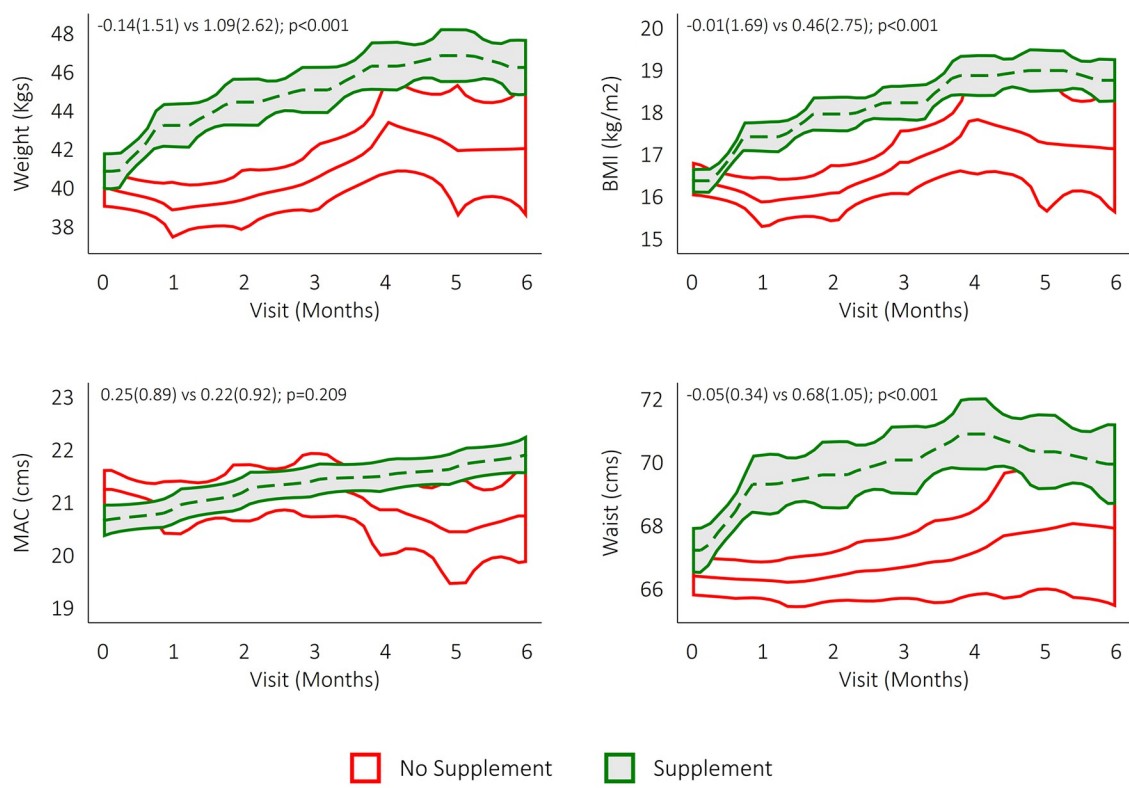

The value indicates the mean change & (%) from the previous visit

**Fig 3. Comparisons over a time period of weight, body mass index, waist circumference, and mid-upper arm circumference in the study participants in the supplement and no supplement group (n = 761).**

sputum conversion [23]. Though the food incentive was acceptable, the fixed timings and supervised consumption in the clinic made the patients uncomfortable and the supplement cost intensive. In our study, the loss to follow-up were significantly lower in the supplement group. The supplement used in this study comprised of locally available uncooked raw ingredients and was supplied to the patient as well as to their family members fortnightly. The patients were allowed to consume it in any form as per their choice and taste. This could have resulted in lower losses for follow-up. Further research studies are warranted on the cost-effectiveness of food incentives and food supplements in TB.

Undernutrition is associated with increased mortality in TB. Studies show that patients who weighed less than 35 kgs and those with moderate (BMI 16.0–16.9 kg/ m$^2$) to severe undernutrition (BMI < 15.9 kg/m$^2$) had a 3.9 times higher chances of death and as early as first four weeks of treatment [2, 25]. It was also seen that for every kg decrease in weight at the start of treatment, odds of survival decreased by 7.5% (P < 0.01) especially among those with drug-resistant TB [26]. Undernutrition also impacts Th1 responses and regulatory T-cell signalling causing a delayed and unregulated immune response [27]. The decreased fat-free mass seen in undernutrition affects the absorption of anti-TB drugs resulting in supratherapeutic levels of drugs [28]. Thus, early rapid correction of undernutrition with food supplements should be explored to avert deaths due to TB.

Studies have shown that there are psychosocial effects of undernutrition in addition to direct biomedical effects on TB treatment outcomes. This emphasizes the need for nutrition interventions and includes HRQOL and mental health-related measures to determine the

**Table 4. Comparisons of WHO—QOL BREF scale and SGRQ over a time period in no supplement and supplement groups by follow-up visit.**

|  | No Supplement (N = 147) | Supplement (N = 614) | p-value |
|---|---|---|---|
| **WHO—QOL BREF** | | | |
| **Physical Domain** | | | |
| Baseline | 35.7 (25.0–46.4) | 25.0 (21.4–32.1) | <0.001 |
| Month 2 | 41.1 (32.1–50.0) | 50.0 (46.4–53.6) | <0.001 |
| Month 6 | 53.6 (48.2–57.1) | 67.9 (60.7–78.6) | <0.001 |
| **Psychological Domain** | | | |
| Baseline | 37.5 (29.2–45.8) | 37.5 (29.2–45.8) | 0.508 |
| Month 2 | 50.0 (41.7–54.2) | 58.3 (54.2–62.5) | <0.001 |
| Month 6 | 58.3 (54.2–66.7) | 79.2 (70.8–79.2) | <0.001 |
| **Social Relationship Domain** | | | |
| Baseline | 41.7 (33.3–50.0) | 50.0 (33.3–58.3) | 0.647 |
| Month 2 | 50.0 (41.7–66.7) | 66.7 (50.0–66.7) | 0.001 |
| Month 6 | 66.7 (58.3–66.7) | 75.0 (66.7–83.3) | <0.001 |
| **Environment Domain** | | | |
| Baseline | 46.9 (37.5–50.0) | 40.6 (34.4–46.9) | 0.015 |
| Month 2 | 46.9 (34.4–53.1) | 50.0 (43.8–53.1) | 0.037 |
| Month 6 | 53.1 (46.9–56.3) | 59.4 (50.0–71.9) | 0.028 |
| **SGRQ Score** | | | |
| **Symptoms** | | | |
| Baseline | 59.3 (44.0–72.5) | 59.3 (48.8–64.0) | 0.676 |
| Month 2 | 43.5 (33.0–51.7) | 33.5 (23.4–41.4) | <0.001 |
| Month 6 | 25.4 (19.2–28.1) | 11.7 (09.4–23.5) | <0.001 |
| **Activity** | | | |
| Baseline | 73.2 (66.5–85.9) | 92.5 (79.0–100) | <0.001 |
| Month 2 | 66.2 (59.5–79.0) | 54.4 (35.8–66.2) | 0.001 |
| Month 6 | 35.5 (24.7–59.9) | 00.0 (00.0–11.2) | <0.001 |
| **Impact** | | | |
| Baseline | 54.5 (28.1–74.7) | 72.9 (58.9–81.1) | <0.001 |
| Month 2 | 43.1 (31.8–57.1) | 25.6 (15.0–37.1) | <0.001 |
| Month 6 | 14.1 (10.4–22.7) | 04.0 (00.0–07.6) | <0.001 |
| **Overall** | | | |
| Baseline | 61.1 (46.1–77.8) | 75.6 (65.0–81.7) | <0.001 |
| Month 2 | 48.9 (40.2–62.2) | 34.7 (24.4–45.0) | <0.001 |
| Month 6 | 25.4 (18.5–33.3) | 05.7 (02.1–10.9) | <0.001 |

Values are given as median (First and Third Quartiles)

Mann-Whitney U test was used at a 5% level of significance

impacts of nutrition interventions [29]. Our study has also shown an improvement in HRQOL and lung health in all the domains in the supplement group.

The strengths of the study include a large cohort with follow-up measurements at several time points. It was done as a phased implementation that allowed comparing those who received food supplements versus those who did not at a given time point. This minimized the biases and enhanced the validity of the study. One of the limitations of the study was that it was done only in one state which could affect the generalisability of the study results. The study was done under field conditions and the risk of refeeding syndrome could not be addressed. The follow up examinations were mainly based on monthly clinical examination,

anthropometry, dietary recall, drug adherence and did not include blood investigations to address the same. Though there were more number of participants with low BMI in the study they were equally distributed in the supplement and no supplement group.

## Conclusion

This study assessed the direct benefit of food supplementation in TB treatment outcomes. Improvement in the nutritional status of the patient can be considered a predictor of treatment success rates. Early food supplementation has a positive impact on the nutritional status. Cost-effective and patient-friendly supplements as an adjunct to treatment are thus needed for favourable treatment outcomes. Further research with a larger sample size and a wider selection of patients in different settings is needed to study the impact of supplements and their implementation in the program.

## Supporting information

**S1 Checklist.** *PLOS ONE* **clinical studies checklist.**
(DOCX)

## Acknowledgments

The authors thank the patients who took part in the study and the field staff of the ICMR-Regional Medical Research Centre, Bhubaneswar, Odisha for engaging the patients in the field conditions, the District TB Officer, and the NTEP Team at Odisha. We extend our gratitude to Dr. Soumya Swaminathan, ex-Director General of the Indian Council of Medical Research who was instrumental in this study. We also thank the Directors, senior scientists, medical officers, and statisticians of ICMR-RMRC and ICMR- NIRT for conducting the study and meticulous data collection.

## Author Contributions

**Conceptualization:** Chandrasekaran Padmapriyadarsini, Beena Thomas, Anand Bang, Soumya Swaminathan.

**Data curation:** Dina Nair.

**Formal analysis:** Kannan Thiruvengadam, Chandrasekaran Padmapriyadarsini.

**Funding acquisition:** Chandrasekaran Padmapriyadarsini, Soumya Swaminathan.

**Investigation:** Gandham Bulliyya.

**Methodology:** Dasarathi Das.

**Project administration:** Amarendra Mahapatra, Chandrasekaran Padmapriyadarsini, Sanghamitra Pati, Dasarathi Das, Anand Bang.

**Resources:** Sanghamitra Pati, Jayeeta Chowdhury, Anand Bang, Soumya Swaminathan.

**Software:** Kannan Thiruvengadam.

**Supervision:** Amarendra Mahapatra, Chandrasekaran Padmapriyadarsini, Beena Thomas, Sanghamitra Pati, Dasarathi Das, Jayeeta Chowdhury.

**Validation:** Dina Nair, Chandrasekaran Padmapriyadarsini.

**Writing – original draft:** Dina Nair, Chandrasekaran Padmapriyadarsini.

**Writing – review & editing:** Amarendra Mahapatra, Kannan Thiruvengadam, Beena Thomas, Sanghamitra Pati, Gandham Bulliyya, Dasarathi Das, Jayeeta Chowdhury, Anand Bang, Soumya Swaminathan.

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
