## [Author Response · Author response to Decision Letter 0]

1 Nov 2023

S. No. Comments Response

Reviewer #1: The study entitled “Effectiveness of Food Supplementation on Treatment Outcomes and Quality of Life in Pulmonary Tuberculosis: Phased Implementation Approach” evaluated the influence of early food supplementation on the TB treatment outcome and quality of life. The following are the suggestions that the authors may consider adding:

1 The discussion is minimal with only 3 references. Several RCTs and other study types have been published with respect to nutritional/food supplementation in TB patients in India and elsewhere. The authors may discuss this with respect to these studies. We have added more references and revised the discussion accordingly

2 The authors have mentioned that food supplementation with rice, finger millet, kandul dal, mustard oil, and sathu was supplied. What is the protein, carbohydrate, minerals, etc. available per gram in these foods? The discussion on this is important since it gives an idea to readers regarding the various macro and micronutrients that were supplied to the patients. This has been now added to the manuscript

3 The authors may discuss the various nutritional deficiencies linked to the risk of developing TB and how the current food supplementation could aim in correcting nutritional deficiencies that could be related to improving the TB outcomes such as smear-negative/culture-negative status etc. We have revised this in the background and discussion section.

4 Since it is mentioned in the introduction that “It is encouraging to note studies showing TB patients who received food supplements during their initial phase of ATT gained weight, had a shorter time to sputum conversion….”, it would be interesting to add information on the influence of food supplementation and an increase in BMI on sputum smear grading in results and discussion. We have added information on the influence of food supplementation and an increase in BMI on sputum smear grading in the results and discussion.

5 What is the average cost incurred for providing food supplementation in the quantities mentioned per family? This may be important for budget planning for programmatic implementation. The cost of the food supplement was budgeted as Rs.1500/- per family. However, we have not done a cost-effectiveness analysis as part of this study.

6 The authors should add more information pertaining to the results and discussion regarding the impact of food supplementation on HRQOL and lung health. We have added more information on the impact of food supplementation on HRQOL and lung health in the results and discussion section. 

7 Typo errors such as of “Hyper Tension” (table 1) can be corrected. We have corrected this in Table 1

Reviewer #2: The topic of providing a nutritional intervention for persons with TB is important, and the results present compelling evidence for the benefits of such an intervention. However, the manuscript itself is not clearly presented and often does not discuss the results and conclusions in a systematic manner. The justification for providing a nutritional intervention is not fully stated, and a reader unfamiliar with the subject could be unsure of the biological relationship between TB and nutrition. Overall, the paper could use an additional review by a copyeditor or writer as there are many instances of less-than-clear English, as well as grammatical and punctuation errors (the majority are not listed below due to their prevalence throughout the manuscript).

1 Line 35: suggest changing “adjuvant” to “adjuvant therapy.” Yes, we have changed it to “Adjuvant therapy”

2 Line 47: this sentence states how many people in the intervention/supplement group experienced a favorable outcome. While the p-value is included, the figure for the control group should also be included in the abstract. The definition of “favorable outcome” should also be included here. The figure for the control group is included in the abstract and the definition of “favorable outcome” is also included.

3 Lines 51-53: these two sentences seem to repeat each other. We have corrected the statement.

4 Lines 65-68: the first sentence of this paragraph describes weight gain during therapy, while the following sentence focuses on malnourishment at time of diagnosis. The two sentences do not logically follow. We have modified the paragraph

5 Lines 72-74: “multidimensional poor” is not explained, and it is not clear how this connects to undernutrition. We have revised the sentence in the background section

6 Lines 83-84: “{3, 3, 3, 3, 3, 4}” is not needed. As suggested by the reviewer, we have deleted the same.

7 Line 85: “Step wedge” should be “stepped wedge” or “stepped-wedge.” We have corrected this terminology and made it similar across the manuscript

8 Introduction: a more detailed biological explanation of the relationship between nutrition and TB would be helpful. A more detailed biological explanation of the relationship between nutrition and TB is added in the introduction

9 Introduction: there is useful information included here, but it is not presented in a completely clear manner. It would be helpful to have a more distinct delineation between: a) the relationship between nutrition and TB; b) the results of previous nutritional interventions; and c) the aims of this study. We have revised the same to have a more distinct delineation between the relationship between nutrition and TB, the results of previous nutritional interventions and the aims of this study.

10 Methods: a justification for why the stepped-wedge approach was used would be useful here. We have added the justification in the methods section.

11 Line 108: why was the food amount provided sufficient for five people for one month? Was only the patient intended to eat the food, or also their family members (and if so, what about households of different sizes)? Family members were not intended, but we provided the same to ensure that the food supplement reaches the patient through this.

12 Line 113: how was medication adherence assessed? Adherence to anti-TB treatment and supplement was ensured by surprise home visits and requisitioning of patient

during monthly visits.

13 Line 133: an explanation of why 74% were male would be helpful here. (Are more TB patients male? Did more men consent to participate compared to women?) Yes, 74% were male as we know that TB is more common in males. 

14 Lines 133-134: with so many having such a low BMI, was the risk of refeeding syndrome addressed? Risk of refeeding syndrome was not addressed. Though there were more number of low BMI in the study it was equally distributed in the supplement and non-supplement group

15 Lines 136-138 present an aIRR, while lines 144-145 present an aRR for the same outcomes. These figures are nearly identical- is there an error here? We have corrected the same.

16 Lines 169-171: information on previous nutritional interventions would be helpful in the Introduction, then could be discussed further in the Discussion section. The information on the previous nutrition interventions is included in the introduction and discussed in the Discussion section

17 Line 179: the psychosocial effects could be discussed at greater length. We have addressed the psychosocial effects and have provided the data in detail in Table – 5. 

18 Lines 184-185: the justification for the stepped-wedge approach presented here is compelling, and should be introduced earlier in the manuscript. Moreover, the comparison at different time points is not discussed within the body of the paper. Yes, we have added the same in the methodology section

19 Lines 187-188: is this a limitation, or an inherent aspect of using a stepped-wedge approach? We do agree that the limitation is an inherent aspect of using a stepped-wedge approach.

20 Line 189: the limitations should be discussed at greater length. We have added the limitations in the discussion section.

21 Discussion: what are the main lessons learned/recommendations for future interventions? We have added the main lessons learned/recommendations for future interventions to the conclusion.

22 Table 1 combines the baseline characteristics and the results. The age, gender, baseline BMI, etc. are appropriate for the demographics table (Table 1), but change in weight, etc. would likely be more appropriate in a results table. Moreover, including p-values in Table 1 may not be relevant. Table 1 compares the supplement and non-supplement and the p-value is provided to confirm the unbiased and equal allocation in both the groups.

23 Table 1: did the analysis address factors such as differential alcohol consumption and sputum smear grading between the groups?

 Yes, we have provided the details of the alcohol consumption, and smear grading between the supplement and non-supplement in Table 1

24 Table 2 and Table 3 would be improved with gridlines.

 Yes, we have added the gridlines as suggested.

25 Figure 1: the different colors are not explained. Yes, we have added a footnote, that the shaded one indicated that the nutrition supplement was rolled out

Reviewer #3: Thank you for the opportunity to review this manuscript investigating the effectiveness of food supplementation on treatment outcomes in patients with TB. Remaining one of the leading infectious causes of morbidity and mortality the research is both timely and relevant to expand the evidence base re supplementary feeding in this context. It was good to note that the researchers also investigated aspects of QoL, which has been lacking in other studies.

Please consider the comments/request for clarity below:

1 Abstract repeat - please correct? Improvement in nutritional status was observed in the supplement group (p<0.001). The supplement group experienced an improvement in nutritional status (p 0.001).

Clarity is needed re the phased approach: rationale for this approach? It was difficult to deduct how the process unfolded, despite the inclusion of Figure 1. The figure gives the impression that Cluster 1 received the supplement for an extended period of time and longer than the other clusters. It is stated that the last cluster only received the supplement for 2 months? Please clarify for the reader, also in the article itself. We have revised the abstract.

In stepped wedge design, the clusters will have intervention periods that vary across and the same has been shown in figure – 1.

2 Please provide more information regarding the food supplement (calories, protein, and other details). Was the idea that the supplement is for use by the whole family? "Food supplement could feed a family of 5 for a month", but provided every 2 weeks? Did the researchers check/ask whether the index patient consumed some of the supplements? More details will provide clarity. We have revised and updated the details. Adherence to anti-TB treatment and supplements was ensured by surprise home visits and requestioning of patients during monthly visits

3 Include references for the 2 questionnaires included in the study and perhaps brief details re the questionnaires. We have added the references in the results and the reference section.

4 Sample: unclear which patients (n=147) did not receive the supplement, and why? The authors could include participant numbers per cluster (those with supplement and those without) to help the reader understand how the study unfolded. The uneven numbers (supplement vs no supplement) are a limitation, but the authors could have realized this at the study design phase. Was it planned as such? Did the researchers employ consecutive sampling? Was there specific inclusion and exclusion criteria? The distribution of participants enrolled in each cluster and the number under the supplement and non-supplement were clearly shown in Figure – 1b

5 Results: why did the researchers only report weight and BMI change at 2 months? Longer time-point data available? The two-month BMI Change is considered the surrogate marker for the TB treatment response. 

6 Discussion: the researchers state the study emphasizes the effect of diet on improving treatment adherence and leading to treatment completion and better outcomes, although not reporting any treatment adherence data under results. How was this measured/evaluated? The discussion is very limited and lacks a critical interrogation of the findings. The authors should also emphasize the novel components of their study and what it adds to the evidence base. We have used the stepped wedge design in a phased manner, as a family-centric approach. The study was done in a resource-constrained area under a programmatic setting.

We have updated the same in the discussion section.

7 Abbreviations should be declared for all tables/figures, even if obvious (e.g. LTFUP). We have declared the abbreviation in the table footnotes for all tables/figures. 

8 The quality of the figures must be drastically improved. We have revised the quality of the figure using the Plos tool

---

## [Decision Letter · Decision Letter 1]

5 Dec 2023

PONE-D-23-12272R1Effectiveness of Food Supplement on Treatment Outcomes and Quality of Life in Pulmonary Tuberculosis: Phased Implementation ApproachPLOS ONE

Dear Dr. Padmapriyadarsini,

Thank you for submitting your manuscript to PLOS ONE. After careful consideration, we feel that it has merit but does not fully meet PLOS ONE’s publication criteria as it currently stands. Therefore, we invite you to submit a revised version of the manuscript that addresses the points raised during the review process.

Please submit your revised manuscript by Jan 19 2024 11:59PM. If you will need significantly more time to complete your revisions, please reply to this message or contact the journal office at plosone@plos.org. Please include the following items when submitting your revised manuscript:A rebuttal letter that responds to each point raised by the academic editor and reviewer(s). You should upload this letter as a separate file labeled 'Response to Reviewers'.A marked-up copy of your manuscript that highlights changes made to the original version. You should upload this as a separate file labeled 'Revised Manuscript with Track Changes'.An unmarked version of your revised paper without tracked changes. You should upload this as a separate file labeled 'Manuscript'.

We look forward to receiving your revised manuscript.

Kind regards,

Frederick Quinn

Academic Editor

PLOS ONE

Reviewers' comments:

Reviewer's Responses to Questions

**Comments to the Author**

1. If the authors have adequately addressed your comments raised in a previous round of review and you feel that this manuscript is now acceptable for publication, you may indicate that here to bypass the “Comments to the Author” section, enter your conflict of interest statement in the “Confidential to Editor” section, and submit your "Accept" recommendation.

Reviewer #2: (No Response)

Reviewer #3: All comments have been addressed

2. Is the manuscript technically sound, and do the data support the conclusions?

Reviewer #2: Partly

Reviewer #3: Yes

3. Has the statistical analysis been performed appropriately and rigorously? 

Reviewer #2: I Don't Know

Reviewer #3: Yes

4. Have the authors made all data underlying the findings in their manuscript fully available?

Reviewer #2: Yes

Reviewer #3: (No Response)

5. Is the manuscript presented in an intelligible fashion and written in standard English?

Reviewer #2: No

Reviewer #3: Yes

6. Review Comments to the Author

Reviewer #2: ***Please see the attached file for reviewer comments with easier-to-read formatting.***

Reviewer #2: The topic of providing a nutritional intervention for persons with TB is important, and the results present compelling evidence for the benefits of such an intervention. However, the manuscript itself is not clearly presented and often does not discuss the results and conclusions in a systematic manner. The justification for providing a nutritional intervention is not fully stated, and a reader unfamiliar with the subject could be unsure of the biological relationship between TB and nutrition. Overall, the paper could use an additional review by a copyeditor or writer as there are many instances of less-than-clear English, as well as grammatical and punctuation errors (the majority are not listed below due to their prevalence throughout the manuscript).

1 Line 35: suggest changing “adjuvant” to “adjuvant therapy.”

Yes, we have changed it to “Adjuvant therapy”

This text has been corrected.

2 Line 47: this sentence states how many people in the intervention/supplement group experienced a favorable outcome. While the p-value is included, the figure for the control group should also be included in the abstract. The definition of “favorable outcome” should also be

included here.

The figure for the control group is included in the abstract and the definition of “favorable outcome” is also included.

This text has been added and increases clarity for the reader. The results of the study are now much easier to understand when reading the Abstract.

3 Lines 51-53: these two sentences seem to repeat each other.

We have corrected the statement.

The text has been corrected.

4 Lines 65-68: the first sentence of this paragraph describes weight gain during therapy, while the following sentence focuses on malnourishment at time of diagnosis. The two sentences do not logically follow.

We have modified the paragraph

These sentences have not been substantially changed, and still do not logically follow.

5 Lines 72-74: “multidimensional poor” is not explained, and it is not clear how this connects to undernutrition.

We have revised the sentence in the background section

This text has been removed.

6 Lines 83-84: “{3, 3, 3, 3, 3, 4}” is not needed.

As suggested by the reviewer, we have deleted the same.

This text has been removed.

7 Line 85: “Step wedge” should be “stepped wedge” or “stepped-wedge.”

We have corrected this terminology and made it similar across the manuscript

This terminology has not been corrected- four different spellings/hyphenations are used in the manuscript.

8 Introduction: a more detailed biological explanation of the relationship between nutrition and TB would be helpful.

A more detailed biological explanation of the relationship between nutrition and TB is added in the introduction

The biological explanation is not included; i.e., why undernutrition is associated with poor treatment outcomes for TB.

9 Introduction: there is useful information included here, but it is not presented in a completely clear manner. It would be helpful to have a more distinct delineation between: a) the relationship between nutrition and TB; b) the results of previous nutritional interventions; and c) the aims of this study.

We have revised the same to have a more distinct delineation between the relationship between nutrition and TB, the results of previous nutritional interventions and the aims of this study.

The Introduction is now better organized.

10 Methods: a justification for why the stepped-wedge approach was used would be useful here.

We have added the justification in the methods section.

The text now states “offered a logical as well as ethical feasibility.” However, a further explanation would be useful.

11 Line 108: why was the food amount provided sufficient for five people for one month? Was only the patient intended to eat the food, or also their family members (and if so, what about households of different sizes)?

Family members were not intended, but we provided the same to ensure that the food supplement reaches the patient through this.

This should be explained within the text.

12 Line 113: how was medication adherence assessed?

Adherence to anti-TB treatment and supplement was ensured by surprise home visits and requisitioning of patient during monthly visits.

This text has been added and increases clarity. Please change “requisitioning” to “requestioning,” or simply “questioning.”

13 Line 133: an explanation of why 74% were male would be helpful here. (Are more TB patients male? Did more men consent to participate compared to women?)

Yes, 74% were male as we know that TB is more common in males.

This answer would be helpful within the paper itself.

14 Lines 133-134: with so many having such a low BMI, was the risk of refeeding syndrome addressed?

Risk of refeeding syndrome was not addressed. Though there were more number of low BMI in the study it was equally distributed in the supplement and nonsupplement group.

Please add this to the Limitations, or add text elsewhere in the paper explaining why refeeding syndrome was not addressed.

15 Lines 136-138 present an aIRR, while lines 144-145 present an aRR for the same outcomes. These figures are nearly identical- is there an error here?

We have corrected the same.

The repeated text has been removed.

16 Lines 169-171: information on previous nutritional interventions would be helpful in the Introduction, then could be discussed further in the Discussion section.

The information on the previous nutrition interventions is included in the introduction and discussed in the Discussion section.

The addition of information on previous nutritional interventions has greatly strengthened the Discussion section.

17 Line 179: the psychosocial effects could be discussed at greater length.

We have addressed the psychosocial effects and have provided the data in detail in Table – 5.

Psychosocial effects are not defined. There is no Table 5 included- does this refer to Table 4?

18 Lines 184-185: the justification for the stepped-wedge approach presented here is compelling, and should be introduced earlier in the manuscript. Moreover, the comparison at different time points is not discussed within the body of the paper.

Yes, we have added the same in the methodology section

More information about the stepped-wedge approach has been added, making the methodology much clearer. However, see comment 10, above.

19 Lines 187-188: is this a limitation, or an inherent aspect of using a stepped-wedge approach?

We do agree that the limitation is an inherent aspect of using a stepped-wedge approach.

The text included here is now clearer.

20 Line 189: the limitations should be discussed at greater length.

We have added the limitations in the discussion section.

Additional limitations have not been added; further discussion of limitations would improve the paper.

21 Discussion: what are the main lessons learned/recommendations for future interventions?

We have added the main lessons learned/recommendations for future interventions to the

conclusion.

Some recommendations for future interventions have been added.

22 Table 1 combines the baseline characteristics and the results. The age, gender, baseline BMI, etc. are appropriate for the demographics table (Table 1), but change in weight, etc. would likely be more appropriate in a results table. Moreover, including p-values in Table 1 may not be relevant.

Table 1 compares the supplement and non-supplement and the p-value is provided to confirm the unbiased and equal allocation in both the groups.

My preference is to include ‘change in weight’ and ‘change in BMI’ in a results table, rather than in Table 1; however, I acknowledge that including these data in Table 1 is a valid stylistic choice.

23 Table 1: did the analysis address factors such as differential alcohol consumption and sputum smear grading between the groups?

Yes, we have provided the details of the alcohol consumption, and smear grading between the

supplement and non-supplement in Table 1

These factors are not discussed within the manuscript. The differential alcohol consumption and sputum smear grading between the groups could have affected the results.

24 Table 2 and Table 3 would be improved with gridlines.

Yes, we have added the gridlines as suggested.

The gridlines have been added.

25 Figure 1: the different colors are not explained.

Yes, we have added a footnote, that the shaded one indicated that the nutrition supplement was rolled out

The footnote has been added to the figure.

Additional comments

Lines 53-54: The meaning of the sentence “The changes from the baseline were significantly higher in the supplement group in terms of nutritional status (p<0.001).” could be expressed more clearly.

Lines 114-118: This information may be better presented in a table.

Lines 191-194: This sentence is unclear.

Discussion: this section is not structured logically, with information on the role of food supplements, the impact of undernutrition on mortality, etc. presented at different times in the Discussion.

Lines 239-241: The text “In this study, more deaths were observed in the supplement group which could be attributed to the fact that most of the participants were extremely underweight in this group” highlights a very important finding/limitation of this study, and should be discussed at greater length. In addition, the following sentence “Thus, early rapid correction of malnutrition with food supplements can prevent deaths due to TB” is not proven by the text that comes before it.

The manuscript’s grammar and sentence structure have significantly improved in this revision. However, there are still areas requiring review.

Reviewer #3: (No Response)

7. PLOS authors have the option to publish the peer review history of their article (what does this mean?). If published, this will include your full peer review and any attached files.

Reviewer #2: No

Reviewer #3: No

---

## [Decision Letter · Decision Letter 0]

21 Jun 2023

PONE-D-23-12272

Effectiveness of Food Supplementation on Treatment Outcomes and Quality of Life in Pulmonary Tuberculosis: Phased Implementation Approach

PLOS ONE

Dear Dr. Padmapriyadarsini,

Thank you for submitting your manuscript to PLOS ONE. After careful consideration, we feel that it has merit but does not fully meet PLOS ONE’s publication criteria as it currently stands. Therefore, we invite you to submit a revised version of the manuscript that addresses the points raised during the review process.

Please submit your revised manuscript by Aug 05 2023 11:59PM. If you will need significantly more time to complete your revisions, please reply to this message or contact the journal office at plosone@plos.org. Please include the following items when submitting your revised manuscript:

We look forward to receiving your revised manuscript.

Kind regards,

Frederick Quinn

Academic Editor

PLOS ONE

Journal Requirements:

"Competing Interest: JC & AB are employees of Tata Trust Foundation of India."

We note that one or more of the authors are employed by a commercial company: Tata Trust Foundation of India.  

Within your Competing Interests Statement, please confirm that this commercial affiliation does not alter your adherence to all PLOS ONE policies on sharing data and materials by including the following statement: ""This does not alter our adherence to  PLOS ONE policies on sharing data and materials.” (as detailed online in our guide for authors http://journals.plos.org/plosone/s/competing-interests) . If this adherence statement is not accurate and  there are restrictions on sharing of data and/or materials, please state these. Please note that we cannot proceed with consideration of your article until this information has been declared.

Reviewers' comments:

Reviewer's Responses to Questions

Comments to the Author

1. Is the manuscript technically sound, and do the data support the conclusions?

Reviewer #1: Yes

Reviewer #2: Yes

Reviewer #3: Yes

2. Has the statistical analysis been performed appropriately and rigorously? 

Reviewer #1: N/A

Reviewer #2: I Don't Know

Reviewer #3: Yes

3. Have the authors made all data underlying the findings in their manuscript fully available?

Reviewer #1: Yes

Reviewer #2: Yes

Reviewer #3: No

4. Is the manuscript presented in an intelligible fashion and written in standard English?

Reviewer #1: Yes

Reviewer #2: No

Reviewer #3: Yes

5. Review Comments to the Author

Reviewer #1: The study entitled “Effectiveness of Food Supplementation on Treatment Outcomes and Quality of Life in Pulmonary Tuberculosis: Phased Implementation Approach” evaluated the influence of early food supplementation on the TB treatment outcome and quality of life. The following are the suggestions that the authors may consider adding:

1. The discussion is minimal with only 3 references. Several RCTs and other study types have been published with respect to nutritional/food supplementation in TB patients in India and elsewhere. The authors may discuss this with respect to these studies.

2. The authors have mentioned that food supplementation with rice, finger millet, kandul dal, mustard oil, and sathu was supplied. What is the protein, carbohydrate, minerals, etc. available per gram in these foods? The discussion on this is important since it gives an idea to readers regarding the various macro and micronutrients that were supplied to the patients.

3. The authors may discuss the various nutritional deficiencies linked to the risk of developing TB and how the current food supplementation could aim in correcting nutritional deficiencies that could be related to improving the TB outcomes such as smear-negative/culture-negative status etc.

4. Since it is mentioned in the introduction that “It is encouraging to note studies showing TB patients who received food supplements during their initial phase of ATT gained weight, had a shorter time to sputum conversion….”, it would be interesting to add information on the influence of food supplementation and an increase in BMI on sputum smear grading in results and discussion.

5. What is the average cost incurred for providing food supplementation in the quantities mentioned per family? This may be important for budget planning for programmatic implementation.

6. The authors should add more information pertaining to the results and discussion regarding the impact of food supplementation on HRQOL and lung health.

7. Typo errors such as of “Hyper Tension” (table 1) can be corrected.

Reviewer #2: The topic of providing a nutritional intervention for persons with TB is important, and the results present compelling evidence for the benefits of such an intervention. However, the manuscript itself is not clearly presented and often does not discuss the results and conclusions in a systematic manner. The justification for providing a nutritional intervention is not fully stated, and a reader unfamiliar with the subject could be unsure of the biological relationship between TB and nutrition. Overall, the paper could use an additional review by a copyeditor or writer as there are many instances of less-than-clear English, as well as grammatical and punctuation errors (the majority are not listed below due to their prevalence throughout the manuscript).

Line 35: suggest changing “adjuvant” to “adjuvant therapy.”

Line 47: this sentence states how many people in the intervention/supplement group experienced a favorable outcome. While the p-value is included, the figure for the control group should also be included in the abstract. The definition of “favorable outcome” should also be included here.

Lines 51-53: these two sentences seem to repeat each other.

Lines 65-68: the first sentence of this paragraph describes weight gain during therapy, while the following sentence focuses on malnourishment at time of diagnosis. The two sentences do not logically follow.

Lines 72-74: “multidimensional poor” is not explained, and it is not clear how this connects to undernutrition.

Lines 83-84: “{3, 3, 3, 3, 3, 4}” is not needed.

Line 85: “Step wedge” should be “stepped wedge” or “stepped-wedge.”

Introduction: a more detailed biological explanation of the relationship between nutrition and TB would be helpful.

Introduction: there is useful information included here, but it is not presented in a completely clear manner. It would be helpful to have a more distinct delineation between: a) the relationship between nutrition and TB; b) the results of previous nutritional interventions; and c) the aims of this study.

Methods: a justification for why the stepped-wedge approach was used would be useful here.

Line 108: why was the food amount provided sufficient for five people for one month? Was only the patient intended to eat the food, or also their family members (and if so, what about households of different sizes)?

Line 113: how was medication adherence assessed?

Line 133: an explanation of why 74% were male would be helpful here. (Are more TB patients male? Did more men consent to participate compared to women?)

Lines 133-134: with so many having such a low BMI, was the risk of refeeding syndrome addressed?

Lines 136-138 present an aIRR, while lines 144-145 present an aRR for the same outcomes. These figures are nearly identical- is there an error here?

Lines 169-171: information on previous nutritional interventions would be helpful in the Introduction, then could be discussed further in the Discussion section.

Line 179: the psychosocial effects could be discussed at greater length.

Lines 184-185: the justification for the stepped-wedge approach presented here is compelling, and should be introduced earlier in the manuscript. Moreover, the comparison at different time points is not discussed within the body of the paper.

Lines 187-188: is this a limitation, or an inherent aspect of used a stepped-wedge approach?

Line 189: the limitations should be discussed at greater length.

Discussion: what are the main lessons learned/recommendations for future interventions?

Table 1 combines the baseline characteristics and the results. The age, gender, baseline BMI, etc. are appropriate for the demographics table (Table 1), but change in weight, etc. would likely be more appropriate in a results table. Moreover, including p-values in Table 1 may not be relevant.

Table 1: did the analysis address factors such as differential alcohol consumption and sputum smear grading between the groups?

Table 2 and Table 3 would be improved with gridlines.

Figure 1: the different colors are not explained.

Reviewer #3: Thank you for the opportunity to review this manuscript investigating the effectiveness of food supplementation on treatment outcomes in patients with TB. Remaining one of the leading infectious causes of morbidity and mortality the research is both timely and relevant to expand the evidence base re supplementary feeding in this context. It was good to note that the researchers also investigating aspects of QoL, which has been lacking in other studies.

Please consider the comments/request for clarity below:

Abstract repeat - please correct? Improvement in nutritional status was observed in the supplement group (p<0.001). The supplement group experienced an improvement in nutritional status (p 0.001).

Clarity is needed re the phased approach: rationale for this approach? It was difficult to deduct how the process unfolded, despite the inclusion of Figure 1. The figure gives the impression that Cluster 1 received the supplement for an extended period of time and longer than the other clusters. It is stated that the last cluster only received the supplement for 2 months? Please clarify for the reader, also in the article itself.

Please provide more information regarding the food supplement (calories, protein and other details). Was the idea that the supplement is for use by the whole family? "Food supplement could feed a family of 5 for a month", but provided every 2 weeks? Did the researchers check/ask whether the index patient consumed some of the supplement? More details will provide clarity.

Include references for the 2 questionnaires included in the study and perhaps brief details re the questionnaires.

Sample: unclear which patients (n=147) did not receive the supplement, and why? The authors could include participant numbers per cluster (those with supplement and those without) to help the reader understand how the study unfolded. The uneven numbers (supplement vs no supplement) is a limitation, but the authors could have realized this at the study design phase? Was it planned as such? Did the researchers employ consecutive sampling? Was there specific inclusion and exclusion criteria?

Results: why did the researchers only report weight and BMI change at 2 months? Longer time-point data available?

Discussion: the researchers state the study emphasizes the effect of diet on improving treatment adherence and leading to treatment completion and better outcomes, although not reporting any treatment adherence data under results? How was this measured/evaluated? The discussion is very limited, and lack a critical interrogation of the findings. The authors should also emphasize the novel components of their study and what it adds to the evidence base.

Abbreviations should be declared for all tables/figures, even if obvious (e.g. LTFUP).

Quality of figures must be drastically improved.

6. PLOS authors have the option to publish the peer review history of their article (what does this mean?). If published, this will include your full peer review and any attached files.

Do you want your identity to be public for this peer review?

 For information about this choice, including consent withdrawal, please see our Privacy Policy.

Reviewer #1: No

Reviewer #2: No

Reviewer #3: No

---

## [Author Response · Author response to Decision Letter 1]

17 Jan 2024

Reviewer #2: The topic of providing a nutritional intervention for persons with TB is important, and the results present compelling evidence for the benefits of such an intervention. However, the manuscript itself is not clearly presented and often does not discuss the results and conclusions in a systematic manner. The justification for providing a nutritional intervention is not fully stated, and a reader unfamiliar with the subject could be unsure of the biological relationship between TB and nutrition. Overall, the paper could use an additional review by a copyeditor or writer as there are many instances of less-than-clear English, as well as grammatical and punctuation errors (the majority are not listed below due to their prevalence throughout the manuscript).

1 Lines 65-68: the first sentence of this paragraph describes weight gain during therapy, while the following sentence focuses on malnourishment at time of diagnosis. The two sentences do not logically follow. These sentences have not been substantially changed, and still do not logically follow.

We have further modified the paragraph and rearranged the sentences so that they flow logically

2 Line 85: “Step wedge” should be “stepped wedge” or “stepped-wedge.” This terminology has not been corrected- four different spellings/hyphenations are used in the manuscript.

We apologize for this. We have now corrected this terminology throughout the manuscript

3 Introduction: a more detailed biological explanation of the relationship between nutrition and TB would be helpful. The biological explanation is not included; i.e., why undernutrition is associated with poor treatment outcomes for TB.

A biological explanation of the relationship between nutrition and TB is now added in the introduction

4 Methods: a justification for why the stepped-wedge approach was used would be useful here. The text now states “offered a logical as well as ethical feasibility.” However, a further explanation would be useful. We have added the justification in the ‘Methods’ section as well as given it here for using this methodology for this study.

“Stepped wedge design (Phased Implementation) is particularly useful when it is not feasible to provide the intervention to every individual/community at once. This design is for evaluating the effectiveness of interventions that have been shown to be efficacious in a more limited, research setting and are now being scaled up to the community level and also useful for evaluating temporal changes in the intervention effect. The intervention is rolled-out sequentially to the trial participants (either as individuals or clusters of individuals) over a number of time periods. The order in which the different individuals or clusters receive the intervention is determined at random and, by the end of the random allocation, all individuals or groups will have received the intervention”

5 Line 108: why was the food amount provided sufficient for five people for one month? Was only the patient intended to eat the food, or also their family members (and if so, what about households of different sizes)? This should be explained within the text.

 Food supplement was not intended for the Family members, but we provided the same to all family members hoping ensure that the food supplement reaches the patient and not become insufficient

6 Line 113: how was medication adherence assessed? Please change “requisitioning” to “requestioning,” or simply “questioning.”

 The spelling has been corrected

7 Line 133: an explanation of why 74% were male would be helpful here. (Are more TB patients male? Did more men consent to participate compared to women?) This answer would be helpful within the paper itself.

 Yes, 74% were male as we know that TB is more common in males and this is now added in the manuscript. 

8 Lines 133-134: with so many having such a low BMI, was the risk of refeeding syndrome addressed? Please add this to the Limitations, or add text elsewhere in the paper explaining why refeeding syndrome was not addressed. Risk of refeeding syndrome was not addressed.

 This has now been added as a limitation of the study.

9 Line 179: the psychosocial effects could be discussed at greater length. 

Psychological and social effects are mentioned in Table 4 and discussed in Discussion section.

10 Lines 184-185: the justification for the stepped-wedge approach presented here is compelling, and should be introduced earlier in the manuscript. Moreover, the comparison at different time points is not discussed within the body of the paper.

 Yes, we have added the same in the methodology section

11 Line 189: the limitations should be discussed at greater length. 

We have added the limitations in the discussion section.

12 Table 1: The differential alcohol consumption and sputum smear grading between the groups could have affected the results.

 We have now added differential alcohol consumption and smear grading status to the text now

All potential factors considered for the model in the articles were tested for collinearity with other independent factors and confounders to the outcome or the groups. After this evaluation, only the appropriate factors were included in the analysis. Alcohol consumption and smear grading of the participant are not significantly associated with each other, although they do significantly differ between the groups. These two covariates were considered and adjusted in all the models and findings. We have discussed the same under the statistical consideration section.

13 Lines 53-54: The meaning of the sentence “The changes from the baseline were significantly higher in the supplement group in terms of nutritional status (p<0.001).” could be expressed more clearly.

 Nutritional status was measured in terms weight, waist circumference, BMI and MAC and is discussed in detail in the results section.

14 The text “In this study, more deaths were observed in the supplement group which could be attributed to the fact that most of the participants were extremely underweight in this group” highlights a very important finding/limitation of this study, and should be discussed at greater length. 

 We have added more information on the same in the discussion section now.

15 Lines 239-241: The text “In this study, more deaths were observed in the supplement group which could be attributed to the fact that most of the participants were extremely underweight in this group” highlights a very important finding/limitation of this study, and should be discussed at greater length. In addition, the following sentence “Thus, early rapid correction of malnutrition with food supplements can prevent deaths due to TB” is not proven by the text that comes before it.

 We have now corrected and added more information on the same.

---

## [Decision Letter · Decision Letter 2]

6 Feb 2024

PONE-D-23-12272R2Effectiveness of Food Supplement on Treatment Outcomes and Quality of Life in Pulmonary Tuberculosis: Phased Implementation ApproachPLOS ONE

Dear Dr. Padmapriyadarsini,

Thank you for submitting your manuscript to PLOS ONE. After careful consideration, we feel that it has merit but does not fully meet PLOS ONE’s publication criteria as it currently stands. Therefore, we invite you to submit a revised version of the manuscript that addresses the points raised during the review process.

Please submit your revised manuscript by Mar 22 2024 11:59PM. If you will need significantly more time to complete your revisions, please reply to this message or contact the journal office at plosone@plos.org. Please include the following items when submitting your revised manuscript:A rebuttal letter that responds to each point raised by the academic editor and reviewer(s). You should upload this letter as a separate file labeled 'Response to Reviewers'.A marked-up copy of your manuscript that highlights changes made to the original version. You should upload this as a separate file labeled 'Revised Manuscript with Track Changes'.An unmarked version of your revised paper without tracked changes. You should upload this as a separate file labeled 'Manuscript'.

We look forward to receiving your revised manuscript.

Kind regards,

Frederick Quinn

Academic Editor

PLOS ONE

Reviewers' comments:

Reviewer's Responses to Questions

**Comments to the Author**

1. If the authors have adequately addressed your comments raised in a previous round of review and you feel that this manuscript is now acceptable for publication, you may indicate that here to bypass the “Comments to the Author” section, enter your conflict of interest statement in the “Confidential to Editor” section, and submit your "Accept" recommendation.

Reviewer #2: (No Response)

2. Is the manuscript technically sound, and do the data support the conclusions?

Reviewer #2: Partly

3. Has the statistical analysis been performed appropriately and rigorously? 

Reviewer #2: I Don't Know

4. Have the authors made all data underlying the findings in their manuscript fully available?

Reviewer #2: No

5. Is the manuscript presented in an intelligible fashion and written in standard English?

Reviewer #2: No

6. Review Comments to the Author

Reviewer #2: Previous comments

Lines 65-68: the first sentence of this paragraph describes weight gain during therapy, while the following sentence focuses on malnourishment at time of diagnosis. The two sentences do not logically follow.

We have modified the paragraph

These sentences have not been substantially changed, and still do not logically follow.

We have further modified the paragraph and rearranged the sentences so that they flow logically

This comment has been addressed.

Line 85: “Step wedge” should be “stepped wedge” or “stepped-wedge.”

We have corrected this terminology and made it similar across the manuscript

This terminology has not been corrected- four different spellings/hyphenations are used in the manuscript.

We apologize for this. We have now corrected this terminology throughout the manuscript

A variety of different spellings are still used in the manuscript.

Introduction: a more detailed biological explanation of the relationship between nutrition and TB would be helpful.

A more detailed biological explanation of the relationship between nutrition and TB is added in the introduction

The biological explanation is not included; i.e., why undernutrition is associated with poor treatment outcomes for TB.

A biological explanation of the relationship between nutrition and TB is now added in the introduction

This has not been added to the manuscript.

Methods: a justification for why the stepped-wedge approach was used would be useful here.

We have added the justification in the methods section.

The text now states “offered a logical as well as ethical feasibility.” However, a further explanation would be useful.

We have added the justification in the ‘Methods’

section as well as given it here for using this methodology for this study.

“Stepped wedge design (Phased Implementation) is particularly useful when it is not feasible to provide the intervention to every individual/community at once. This design is for evaluating the effectiveness of interventions that have been shown to be efficacious in a more limited, research setting and are now being scaled up to the community level and also useful for evaluating temporal changes in the intervention effect.The intervention is rolled-out sequentially to the trial

participants (either as individuals or clusters of individuals) over a number of time periods. The order in which the different individuals or clusters receive the intervention is determined at random and, by the end of the random allocation, all individuals or groups will have received the intervention”

This has not been added to the manuscript.

Line 108: why was the food amount provided sufficient for five people for one month? Was only the patient intended to eat the food, or also their family members (and if so, what about households of different sizes)?

Family members were not intended, but we provided the same to ensure that the food supplement reaches the patient through this.

This should be explained within the text.

Food supplement was not intended for the Family members, but we provided the same to all family members hoping ensure that the food supplement reaches the patient and not become insufficient.

This has not been added to the manuscript.

Line 113: how was medication adherence assessed?

Adherence to anti-TB treatment and supplement was ensured by surprise home visits and requisitioning of patient during monthly visits.

This text has been added and increases clarity. Please change “requisitioning” to “requestioning,” or simply “questioning.”

The spelling has been corrected

Comment addressed.

Line 133: an explanation of why 74% were male would be helpful here. (Are more TB patients male? Did more men consent to participate compared to women?)

Yes, 74% were male as we know that TB is more common in males.

This answer would be helpful within the paper itself.

this is now added in the manuscript.

This has not been added to the manuscript.

Lines 133-134: with so many having such a low BMI, was the risk of refeeding syndrome addressed?

Risk of refeeding syndrome was not addressed. Though there were more number of low BMI in the study it was equally distributed in the supplement and nonsupplement group.

Please add this to the Limitations, or add text elsewhere in the paper explaining why refeeding syndrome was not addressed.

Risk of refeeding syndrome was not addressed. This has now been added as a limitation of the study.

This has not been added to the manuscript.

Line 179: the psychosocial effects could be discussed at greater length.

We have addressed the psychosocial effects and have provided the data in detail in Table – 5.

Psychosocial effects are not defined. There is no Table 5 included- does this refer to Table 4?

Psychological and social effects are mentioned in Table 4 nd discussed in Discussion section.

The psychosocial effects have not yet been defined.

Lines 184-185: the justification for the stepped-wedge approach presented here is compelling, and should be introduced earlier in the manuscript. Moreover, the comparison at different time points is not discussed within the body of the paper.

Yes, we have added the same in the methodology section

More information about the stepped-wedge approach has been added, making the methodology much clearer. However, see comment 10, above.

Yes, we have added the same in the methodology section

Comment addressed

Table 1: did the analysis address factors such as differential alcohol consumption and sputum smear grading between the groups?

Yes, we have provided the details of the alcohol consumption, and smear grading between the

supplement and non-supplement in Table 1

These factors are not discussed within the manuscript. The differential alcohol consumption and sputum smear grading between the groups could have affected the results.

We have now added differential alcohol consumption and smear grading status to the text now

All potential factors considered for the model in the articles were tested for collinearity with other independent factors and confounders to the outcome or the groups. After this evaluation, only the appropriate factors were included in the analysis. Alcohol consumption and smear grading of the participant are not significantly associated with each other, although they do significantly differ between the groups. These two covariates were considered and

adjusted in all the models and findings. We have discussed the same under the statistical consideration section.

This text has not been added to the manuscript.

Additional comments

Lines 53-54: The meaning of the sentence “The changes from the baseline were significantly higher in the supplement group in terms of nutritional status (p<0.001).” could be expressed more clearly.

Nutritional status was measured in terms weight, waist circumference, BMI and MAC and is discussed in detail in the results section.

Comment addressed.

Lines 114-118: This information may be better presented in a table.

Not addressed.

Lines 191-194: This sentence is unclear.

Not addressed.

Discussion: this section is not structured logically, with information on the role of food supplements, the impact of undernutrition on mortality, etc. presented at different times in the Discussion.

Not addressed.

Lines 239-241: The text “In this study, more deaths were observed in the supplement group which could be attributed to the fact that most of the participants were extremely underweight in this group” highlights a very important finding/limitation of this study, and should be discussed at greater length. In addition, the following sentence “Thus, early rapid correction of malnutrition with food supplements can prevent deaths due to TB” is not proven by the text that comes before it.

We have added more information on the same in the discussion section now. We have now corrected and added more information on the same.

This section still contains logical inconsistencies/lack of clarity. “More deaths in the supplement group” would appear to prove the opposite of the point that “food supplements can prevent deaths due to TB.” At a minimum, this statement needs to be “may” or “might”, but more emphasis should be placed on the limitation that most of the participants in the supplement group were extremely underweight and had increased mortality. The authors could also instead highlight the increased favorable outcomes in the supplement group, or the significantly higher increase in BMI.

7. PLOS authors have the option to publish the peer review history of their article (what does this mean?). If published, this will include your full peer review and any attached files.

Reviewer #2: No

---

## [Author Response · Author response to Decision Letter 2]

14 Feb 2024

We have responded to specific reviewer and editor comments

---

## [Decision Letter · Decision Letter 3]

19 Mar 2024

PONE-D-23-12272R3Effectiveness of Food Supplement on Treatment Outcomes and Quality of Life in Pulmonary Tuberculosis: Phased Implementation ApproachPLOS ONE

Dear Dr. Padmapriyadarsini,

Thank you for submitting your manuscript to PLOS ONE. After careful consideration, we feel that it has merit but does not fully meet PLOS ONE’s publication criteria as it currently stands. Therefore, we invite you to submit a revised version of the manuscript that addresses the points raised during the review process.

Please submit your revised manuscript by May 03 2024 11:59PM. If you will need significantly more time to complete your revisions, please reply to this message or contact the journal office at plosone@plos.org. Please include the following items when submitting your revised manuscript:A rebuttal letter that responds to each point raised by the academic editor and reviewer(s). You should upload this letter as a separate file labeled 'Response to Reviewers'.A marked-up copy of your manuscript that highlights changes made to the original version. You should upload this as a separate file labeled 'Revised Manuscript with Track Changes'.An unmarked version of your revised paper without tracked changes. You should upload this as a separate file labeled 'Manuscript'.If applicable, we recommend that you deposit your laboratory protocols in protocols.io to enhance the reproducibility of your results. Protocols.io assigns your protocol its own identifier (DOI) so that it can be cited independently in the future. For instructions see: https://journals.plos.org/plosone/s/submission-guidelines#loc-laboratory-protocols. Additionally, PLOS ONE offers an option for publishing peer-reviewed Lab Protocol articles, which describe protocols hosted on protocols.io. Read more information on sharing protocols at https://plos.org/protocols?utm_medium=editorial-email&utm_source=authorletters&utm_campaign=protocols.

We look forward to receiving your revised manuscript.

Kind regards,

Frederick Quinn

Academic Editor

PLOS ONE

Journal Requirements:

Reviewers' comments:

Reviewer's Responses to Questions

**Comments to the Author**

1. If the authors have adequately addressed your comments raised in a previous round of review and you feel that this manuscript is now acceptable for publication, you may indicate that here to bypass the “Comments to the Author” section, enter your conflict of interest statement in the “Confidential to Editor” section, and submit your "Accept" recommendation.

Reviewer #3: All comments have been addressed

Reviewer #4: All comments have been addressed

2. Is the manuscript technically sound, and do the data support the conclusions?

Reviewer #3: Yes

Reviewer #4: Yes

3. Has the statistical analysis been performed appropriately and rigorously? 

Reviewer #3: Yes

Reviewer #4: Yes

4. Have the authors made all data underlying the findings in their manuscript fully available?

Reviewer #3: No

Reviewer #4: No

5. Is the manuscript presented in an intelligible fashion and written in standard English?

Reviewer #3: Yes

Reviewer #4: Yes

6. Review Comments to the Author

Reviewer #3: (No Response)

Reviewer #4: This step-wedge cluster-randomized study of nutritional support for PWTB is a valuable addition to the literature as it seeks to quantify the impact of in-kind nutritional support on treatment outcomes. The authors are to be lauded for conducting this critical research. It is a timely topic too in the aftermath of the landmark RATIONS study.

I believe this article should be accepted with some minor revisions.

Major points:

1. The authors should strongly consider providing more detailed descriptive statistics regarding the outcomes in each arm. Given the interest in understanding the mechanism by which the nutritional intervention improves outcomes, I would expect that they would have provided a table with the number of participants having 1. Treatment success or cure 2. Death 3. Failure 4. Loss-to-follow up in the nutritional supplementation and the control group. The authors have nicely analyzed differences in outcomes using regression models, but raw numbers in addition would help readers better understand what drove the improved outcomes. This has been done well previously in an earlier study from Bengal (PMID: 28042591)

2. Given that nutrition is a gendered topic with women routinely having worse nutritional intake than men, it would also be helpful to assess differences in men and women in terms of weight gain and providing that at least as a supplementary figure. There is a paucity of such excellent nutritional supplementation studies in PWTB so these data are very valuable and informative to the community of researchers and scientists. This has been done well in a recent nutritional supplementation study in Madhya Pradesh (https://doi.org/10.1016/j.cegh.2021.100782)

3. I was unsure why the authors selected the BMI cutoffs that they did. The more traditional BMI cutoffs, as defined by the WHO, are BMI <16 kg/m2 for severe undernutrition and 16-18.5 would be considered mild-moderate. 18.5-24.9 is considered “normal,” as the authors have suggested, but a BMI >23 is generally considered overweight among South Asians. I recommend adjusting BMI values to more conventional thresholds to facilitate comparability with large nutrition-TB studies such as the recent RePORT India publication (PMID: 36424864). If not, please provide a justification.  

4. Please state more clearly how long your intervention was. I inferred it was just 1 month long from the text, but this should be stated very clearly. This is critical to allow comparison. In general, I think the presentation of the nutritional intervention could be perhaps improved. I recommend adapting the table to resemble the informative table 1 from this protocol paper for an ongoing nutritional intervention (PMID: 34641820)

5. Please comment on the socioeconomic status of the participants of this study and if possible include a deprivation measure such as the kuppuswamy scale (PMC6618222) or at least income in your analysis if possible. Poverty is a driver of unfavorable outcomes and can be an unmeasured confounder. If it is not possible to include it in this analysis, at least describe the general economic status of the population (e.g. is it comparable to the extremely impoverished population of the RATIONS study?) and acknowledge this as a study limitation.

Minor points:

1. Please use person-first language. Instead of TB patient, please use person with TB (PWTB) as that is less stigmatizing

2. Sentence seems incomplete: “…500gms of “Sathu” (flour made from 112 groundnut, wheat, flat rice, and chickpea which contained the following - protein 19gm, fat 113 6gms, carbohydrates 71gm, calcium 200mg, energy 413.31kcal, Vit A 300IU, Vit C 300mg 114 and Vitamin D100IU.”

3. I was surprised to see two of the three largest TB outcome studies from India excluded from the paper’s discussion in favor of smaller studies from non-Indian populations: (PMID: 24205052 and PMID: 36424864)

7. PLOS authors have the option to publish the peer review history of their article (what does this mean?). If published, this will include your full peer review and any attached files.

Reviewer #3: No

Reviewer #4: No

---

## [Author Response · Author response to Decision Letter 3]

30 Apr 2024

PONE-D-23-12272 - Response to Reviewers comments (R4)

1.The authors should strongly consider providing more detailed descriptive statistics regarding the outcomes in each arm. Given the interest in understanding the mechanism by which the nutritional intervention improves outcomes, I would expect that they would have provided a table with the number of participants having 1. Treatment success or cure 2. Death 3. Failure 4. Loss-to-follow up in the nutritional supplementation and the control group. The authors have nicely analyzed differences in outcomes using regression models, but raw numbers in addition would help readers better understand what drove the improved outcomes. This has been done well previously in an earlier study from Bengal (PMID: 28042591)

Response: Thank you. As suggested, a detailed descriptive statistic regarding the outcomes in each arm is mentioned in the result section lines 181-186 (treatment outcomes) and in detail in the Table 2 (highlighted) in the manuscript.

2. Given that nutrition is a gendered topic with women routinely having worse nutritional intake than men, it would also be helpful to assess differences in men and women in terms of weight gain and providing that at least as a supplementary figure. There is a paucity of such excellent nutritional supplementation studies in PWTB so these data are very valuable and informative to the community of researchers and scientists. This has been done well in a recent nutritional supplementation study in Madhya Pradesh (https://doi.org/10.1016/j.cegh.2021.100782) 

Response: Yes, we also agree that nutrition is a gendered topic with women routinely having worse nutritional intake than men and towards this we did a subgroup analysis to assess the differences in men and women in terms of weight gain. The comparison of weight gain between male and female participants in our study did not reveal any significant differences or associations. Hence, we focused on the differences in weight gain during the second and sixth months, as well as its association with outcomes for each gender separately. In the multivariate analysis we have adjusted for the gender to ensure our findings.

3. I was unsure why the authors selected the BMI cutoffs that they did. The more traditional BMI cutoffs, as defined by the WHO, are BMI <16 kg/m2 for severe undernutrition and 16-18.5 would be considered mild-moderate. 18.5-24.9 is considered “normal,” as the authors have suggested, but a BMI >23 is generally considered overweight among South Asians. I recommend adjusting BMI values to more conventional thresholds to facilitate comparability with large nutrition-TB studies such as the recent RePORT India publication (PMID: 36424864). If not, please provide a justification. 

Response: Thank you, for the comment. We had taken the reference from Guidance Document - Nutritional Care & Support for TB patients in India 2017.We have modified the table 1 and also the reference in the manuscript.

4. Please state more clearly how long your intervention was. I inferred it was just 1 month long from the text, but this should be stated very clearly. This is critical to allow comparison. In general, I think the presentation of the nutritional intervention could be perhaps improved. I recommend adapting the table to resemble the informative table 1 from this protocol paper for an ongoing nutritional intervention (PMID: 34641820),

Response: Thank you for the suggestion. The presentation of the nutritional intervention is modified in the box. Also, it is revised in the manuscript

5. Please comment on the socioeconomic status of the participants of this study and if possible, include a deprivation measure such as the kuppuswamy scale (PMC6618222) or at least income in your analysis if possible. Poverty is a driver of unfavorable outcomes and can be an unmeasured confounder. If it is not possible to include it in this analysis, at least describe the general economic status of the population (e.g. is it comparable to the extremely impoverished population of the RATIONS study?) and acknowledge this as a study limitation.

Response: Yes, we also agree that poverty is a driver of unfavorable outcomes and can be an unmeasured confounder. But, the comparison of income between the groups did not show any significant differences or associations either within the groups or across the outcomes. (Table 1) 

Minor points:

1. Please use person-first language. Instead of TB patient, please use person with TB (PWTB) 

 as that is less stigmatizing

 Response: We have revised the sentence in the text (highlighted).

2. Sentence seems incomplete: “…500gms of “Sathu” (flour made from 112 groundnut, wheat, 

 flat rice, and chickpea which contained the following - protein 19gm, fat 113 6gms, 

 carbohydrates 71gm, calcium 200mg, energy 413.31kcal, Vit A 300IU, Vit C 300mg 114 and 

 Vitamin D100IU.”

 Response: We have revised the sentence in the text (highlighted).

3. I was surprised to see two of the three largest TB outcome studies from India excluded from 

 the paper’s discussion in favor of smaller studies from non-Indian populations: 

 (PMID: 24205052 and PMID: 36424864)

 Response: We have included PMID 24205052 in our discussion (reference no.11) 

 (highlighted). PMID 36424864 was published after we submitted the manuscript to the 

 journal. Hence, we were not able to reference the paper in our manuscript.

---

## [Decision Letter · Decision Letter 4]

23 May 2024

PONE-D-23-12272R4Effectiveness of Food Supplement on Treatment Outcomes and Quality of Life in Pulmonary Tuberculosis: Phased Implementation ApproachPLOS ONE

Dear Dr. Padmapriyadarsini,

Thank you for submitting your manuscript to PLOS ONE. After careful consideration, we feel that it has merit but does not fully meet PLOS ONE’s publication criteria as it currently stands. Therefore, we invite you to submit a revised version of the manuscript that addresses the points raised during the review process.

Please submit your revised manuscript by Jul 07 2024 11:59PM. If you will need significantly more time to complete your revisions, please reply to this message or contact the journal office at plosone@plos.org. Please include the following items when submitting your revised manuscript:A rebuttal letter that responds to each point raised by the academic editor and reviewer(s). You should upload this letter as a separate file labeled 'Response to Reviewers'.A marked-up copy of your manuscript that highlights changes made to the original version. You should upload this as a separate file labeled 'Revised Manuscript with Track Changes'.An unmarked version of your revised paper without tracked changes. You should upload this as a separate file labeled 'Manuscript'.If applicable, we recommend that you deposit your laboratory protocols in protocols.io to enhance the reproducibility of your results. Protocols.io assigns your protocol its own identifier (DOI) so that it can be cited independently in the future. For instructions see: https://journals.plos.org/plosone/s/submission-guidelines#loc-laboratory-protocols. Additionally, PLOS ONE offers an option for publishing peer-reviewed Lab Protocol articles, which describe protocols hosted on protocols.io. Read more information on sharing protocols at https://plos.org/protocols?utm_medium=editorial-email&utm_source=authorletters&utm_campaign=protocols.

We look forward to receiving your revised manuscript.

Kind regards,

Frederick Quinn

Academic Editor

PLOS ONE

Journal Requirements:

Reviewers' comments:

Reviewer's Responses to Questions

**Comments to the Author**

1. If the authors have adequately addressed your comments raised in a previous round of review and you feel that this manuscript is now acceptable for publication, you may indicate that here to bypass the “Comments to the Author” section, enter your conflict of interest statement in the “Confidential to Editor” section, and submit your "Accept" recommendation.

Reviewer #4: (No Response)

2. Is the manuscript technically sound, and do the data support the conclusions?

Reviewer #4: Yes

3. Has the statistical analysis been performed appropriately and rigorously? 

Reviewer #4: Yes

4. Have the authors made all data underlying the findings in their manuscript fully available?

Reviewer #4: No

5. Is the manuscript presented in an intelligible fashion and written in standard English?

Reviewer #4: Yes

6. Review Comments to the Author

Reviewer #4: Thank you for your excellent response. The manuscript is essentially ready. I think it just needs two things:

1. After reading the revised manuscipt, I am still confused what the duration of nutritional support was: "The total quantity provided supported a family of five for one month and cost Rs.1500/- per month." Does this mean that that they received the supplementation just for one month or throughout the treatment at a cost of 1500 rupees per month.

Please state specifically that the supplementation was provided for x months.

2. As the authors know, income is an imperfect marker of socioeconomic status, but it is better than nothing. In their response to the reviewer, they stated no differences between the control and the intervention group in income, but I don't see this in their descriptive statistics table. This should be added. And if there is a significant difference, it should be included in the multivariable model.

7. PLOS authors have the option to publish the peer review history of their article (what does this mean?). If published, this will include your full peer review and any attached files.

Reviewer #4: No

---

## [Author Response · Author response to Decision Letter 4]

27 May 2024

PONE-D-23-12272 - Response to Reviewers comments (R5)

Reviewer #4

1. After reading the revised manuscript, I am still confused what the duration of nutritional support was: "The total quantity provided supported a family of five for one month and cost Rs.1500/- per month." Does this mean that that they received the supplementation just for one month or throughout the treatment at a cost of 1500 rupees per month.

Please state specifically that the supplementation was provided for x months.

Response: Thank you for the comment. The food supplement was supplied to the index patient as fortnightly packs at the Directly observed treatment short course (DOTS) Center by the study staff till the end of his /her treatment period. We have now stated this specifically in the revised version of the manuscript. (Line 117-119 -highlighted)

2. As the authors know, income is an imperfect marker of socioeconomic status, but it is better than nothing. In their response to the reviewer, they stated no differences between the control and the intervention group in income, but I don't see this in their descriptive statistics table. This should be added. And if there is a significant difference, it should be included in the multivariable model.

Response: Thank you, we also agree to the comment and we have added the same in the descriptive statistics table. (Table 1-highlighted)

---

## [Decision Letter · Decision Letter 5]

6 Jun 2024

Effectiveness of Food Supplement on Treatment Outcomes and Quality of Life in Pulmonary Tuberculosis: Phased Implementation Approach

PONE-D-23-12272R5

Dear Dr. Padmapriyadarsini,

We’re pleased to inform you that your manuscript has been judged scientifically suitable for publication and will be formally accepted for publication once it meets all outstanding technical requirements.

Kind regards,

Frederick Quinn

Academic Editor

PLOS ONE

Additional Editor Comments (optional):

Reviewers' comments:

Reviewer's Responses to Questions

**Comments to the Author**

1. If the authors have adequately addressed your comments raised in a previous round of review and you feel that this manuscript is now acceptable for publication, you may indicate that here to bypass the “Comments to the Author” section, enter your conflict of interest statement in the “Confidential to Editor” section, and submit your "Accept" recommendation.

Reviewer #4: All comments have been addressed

2. Is the manuscript technically sound, and do the data support the conclusions?

Reviewer #4: (No Response)

3. Has the statistical analysis been performed appropriately and rigorously? 

Reviewer #4: Yes

4. Have the authors made all data underlying the findings in their manuscript fully available?

Reviewer #4: Yes

5. Is the manuscript presented in an intelligible fashion and written in standard English?

Reviewer #4: Yes

6. Review Comments to the Author

Reviewer #4: (No Response)

7. PLOS authors have the option to publish the peer review history of their article (what does this mean?). If published, this will include your full peer review and any attached files.

Reviewer #4: **Yes: **Pranay Sinha, MD, SM

---

## [Editor Report · Acceptance letter]

8 Jul 2024

PONE-D-23-12272R5 

PLOS ONE

Dear Dr. Padmapriyadarsini, 

I'm pleased to inform you that your manuscript has been deemed suitable for publication in PLOS ONE. Congratulations! Your manuscript is now being handed over to our production team.

Kind regards, 

on behalf of

Dr. Frederick Quinn 

Academic Editor

PLOS ONE